# Space-Time Local Embeddings

**Ke Sun**[1*]    **Jun Wang**[2]    **Alexandros Kalousis**[3,1]    **Stéphane Marchand-Maillet**[1]
[1] Viper Group, Computer Vision and Multimedia Laboratory, University of Geneva
`sunk.edu@gmail.com`, `Stephane.Marchand-Maillet@unige.ch`, and [2] Expedia,
Switzerland, `jwang1@expedia.com`, and [3] Business Informatics Department, University
of Applied Sciences, Western Switzerland, `Alexandros.Kalousis@hesge.ch`

## Abstract

Space-time is a profound concept in physics. This concept was shown to be useful for dimensionality reduction. We present basic definitions with interesting counter-intuitions. We give theoretical propositions to show that space-time is a more powerful representation than Euclidean space. We apply this concept to manifold learning for preserving local information. Empirical results on non-metric datasets show that more information can be preserved in space-time.

## 1   Introduction

As a simple and intuitive representation, the Euclidean space $\Re^d$ has been widely used in various learning tasks. In dimensionality reduction, $n$ given high-dimensional points in $\Re^D$, or their pairwise (dis-)similarities, are usually represented as a corresponding set of points in $\Re^d$ ($d < D$).

The representation power of $\Re^d$ is limited. Some of its limitations are listed next. ① The maximum number of points which can share a common nearest neighbor is limited (2 for $\Re$; 5 for $\Re^2$) [1, 2], while such *centralized structures* do exist in real data. ② $\Re^d$ can at most embed $(d + 1)$ points with uniform pair-wise similarities. It is hard to model pair-wise relationships with less variance. ③ Even if $d$ is large enough, $\Re^d$ as a metric space must satisfy the triangle inequality, and therefore must admit transitive similarities [2], meaning that a neighbor's neighbor should also be nearby. Such relationships can be violated on real data, e.g. social networks. ④ The Gram matrix of $n$ real vectors must be positive semi-definite (p. s. d.). Therefore $\Re^d$ cannot faithfully represent the negative eigen-spectrum of input similarities, which was discovered to be meaningful [3].

To tackle the above limitations of Euclidean embeddings, a commonly-used method is to impose a statistical mixture model. Each embedding point is a random point on several candidate locations w. r. t. some mixture weights. These candidate locations can be in the same $\Re^d$ [4]. This allows an embedding point to jump across a long distance through a "statistical worm-hole". Or, they can be in $m$ independent $\Re^d$'s [2, 5], resulting in $m$ different views of the input data.

Another approach beyond Euclidean embeddings is to change the embedding destination to a curved space $\mathcal{M}^d$. This $\mathcal{M}^d$ can be a Riemannian manifold [6] with a positive definite metric, or equivalently, a curved surface embedded in a Euclidean space [7, 8]. To learn such an embedding requires a closed-form expression of the distance measure. This $\mathcal{M}^d$ can also be semi-Riemannian [9] with an indefinite metric. This semi-Riemannian representation, under the names "pseudo-Euclidean space", "Minkowski space", or more conveniently, "space-time", was shown [3, 7, 10–12] to be a powerful representation for non-metric datasets. In these works, an embedding is obtained through a spectral decomposition of a "pseudo-Gram" matrix, which is computed based on some input data.

On the other hand, manifold learning methods [4, 13, 14] are capable of *learning* a p. s. d. kernel Gram matrix, that encapsulates useful information into a narrow band of its eigen-spectrum.

---

[*]Corresponding author

Usually, local neighborhood information is more strongly preserved as compared to non-local information [4, 15], so that the input information is *unfolded* in a non-linear manner to achieve the desired compactness.

The present work advocates the space-time representation. Section 2 introduces the basic concepts. Section 3 gives several simple propositions that describe the representation power of space-time. As novel contributions, section 4 applies the space-time representation to manifold learning. Section 5 shows that using the same number of parameters, more information can be preserved by such embeddings as compared to Euclidean embeddings. This leads to new data visualization techniques. Section 6 concludes and discusses possible extensions.

## 2 Space-time

The fundamental measurements in geometry are established by the concept of a metric [6]. Intuitively, it is a locally- or globally-defined inner product. The metric of a Euclidean space $\Re^d$ is everywhere identity. The inner product between any two vectors $\boldsymbol{y}_1$ and $\boldsymbol{y}_2$ is $\langle \boldsymbol{y}_1, \boldsymbol{y}_2 \rangle = \boldsymbol{y}_1^T \boldsymbol{I}_d \boldsymbol{y}_2$, where $\boldsymbol{I}_d$ is the $d \times d$ identity matrix. A *space-time* $\Re^{d_s, d_t}$ is a $(d_s + d_t)$-dimensional real vector space, where $d_s \geq 0$, $d_t \geq 0$, and the metric is

$$\boldsymbol{M} = \begin{bmatrix} \boldsymbol{I}_{d_s} & \boldsymbol{0} \\ \boldsymbol{0} & -\boldsymbol{I}_{d_t} \end{bmatrix}. \tag{1}$$

This metric is not trivial. It is semi-Riemannian with a background in physics [9]. A point in $\Re^{d_s, d_t}$ is called an *event*, denoted by $\boldsymbol{y} = (y^1, \ldots, y^{d_s}, y^{d_s+1}, \ldots, y^{d_s+d_t})^T$. The first $d_s$ dimensions are *space-like*, where the measurements are exactly the same as in a Euclidean space. The last $d_t$ dimensions are *time-like*, which cause counter-intuitions. In accordance to the metric $\boldsymbol{M}$ in eq. (1),

$$\forall \boldsymbol{y}_1, \boldsymbol{y}_2 \in \Re^{d_s, d_t}, \quad \langle \boldsymbol{y}_1, \boldsymbol{y}_2 \rangle = \sum_{l=1}^{d_s} y_1^l y_2^l - \sum_{l=d_s+1}^{d_s+d_t} y_1^l y_2^l. \tag{2}$$

In analogy to using inner products to define distances, the following definition gives a dissimilarity measure between two events in $\Re^{d_s, d_t}$.

**Definition 1.** *The* space-time interval, *or shortly* interval, *between any two events $\boldsymbol{y}_1$ and $\boldsymbol{y}_2$ is*

$$c(\boldsymbol{y}_1, \boldsymbol{y}_2) = \langle \boldsymbol{y}_1, \boldsymbol{y}_1 \rangle + \langle \boldsymbol{y}_2, \boldsymbol{y}_2 \rangle - 2\langle \boldsymbol{y}_1, \boldsymbol{y}_2 \rangle = \sum_{l=1}^{d_s} (y_1^l - y_2^l)^2 - \sum_{l=d_s+1}^{d_s+d_t} (y_1^l - y_2^l)^2. \tag{3}$$

The space-time interval $c(\boldsymbol{y}_1, \boldsymbol{y}_2)$ can be positive, zero or negative. With respect to a reference point $\boldsymbol{y}_0 \in \Re^{d_s, d_t}$, the set $\{\boldsymbol{y} : c(\boldsymbol{y}, \boldsymbol{y}_0) = 0\}$ is called a *light cone*. Figure 1a shows a light cone in $\Re^{2,1}$. Within the light cone, $c(\boldsymbol{y}, \boldsymbol{y}_0) < 0$, i. e., negative interval occurs; outside the light cone, $c(\boldsymbol{y}, \boldsymbol{y}_0) > 0$. The following *counter-intuitions* help to establish the concept of space-time.

A low-dimensional $\Re^{d_s, d_t}$ can accommodate an arbitrarily large number of events sharing a common nearest neighbor. In $\Re^{2,1}$, let $\boldsymbol{A} = (0, 0, 1)$, and put $\{\boldsymbol{B}_1, \boldsymbol{B}_2, \ldots,\}$ evenly on the circle $\{(y^1, y^2, 0) : (y^1)^2 + (y^2)^2 = 1\}$ at time 0. Then, $\boldsymbol{A}$ is the unique nearest neighbor of $\boldsymbol{B}_1, \boldsymbol{B}_2, \ldots$.

A low-dimensional $\Re^{d_s, d_t}$ can represent uniform pair-wise similarities between an arbitrarily large number of points. In $\Re^{1,1}$, the similarities within $\{\boldsymbol{A}_i : \boldsymbol{A}_i = (i, i)\}_{i=1}^n$ are uniform.

In $\Re^{d_s, d_t}$, the triangle inequality is not necessarily satisfied. In $\Re^{2,1}$, let $\boldsymbol{A} = (-1, 0, 0)$, $\boldsymbol{B} = (0, 0, 1)$, $\boldsymbol{C} = (1, 0, 0)$. Then $c(\boldsymbol{A}, \boldsymbol{C}) > c(\boldsymbol{A}, \boldsymbol{B}) + c(\boldsymbol{B}, \boldsymbol{C})$. The trick is that, as $\boldsymbol{B}$'s absolute time value increases, its intervals with all events at time 0 are shrinking. Correspondingly, similarity measures in $\Re^{d_s, d_t}$ can be *non-transitive*. The fact that $\boldsymbol{B}$ is similar to $\boldsymbol{A}$ and $\boldsymbol{C}$ independently does not necessarily mean that $\boldsymbol{A}$ and $\boldsymbol{C}$ are similar.

A neighborhood of $\boldsymbol{y}_0 \in \Re^{2,1}$ is $\{(y^1, y^2, y^3) : (y^1 - y_0^1)^2 + (y^2 - y_0^2)^2 - (y^3 - y_0^3)^2 \leq \epsilon\}$, where $\epsilon \in \Re$. This hyperboloid has infinite "volume", no matter how small $\epsilon$ is. Comparatively, a neighborhood in $\Re^d$ is much narrower, with an exponentially shrinking volume as its radius decreases.

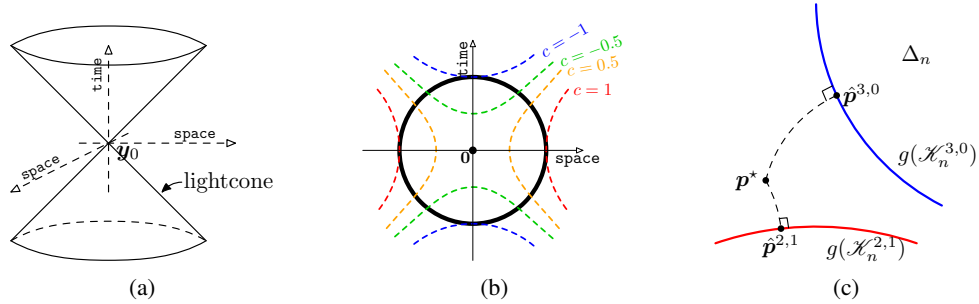

Figure 1: (a) A space-time; (b) A space-time "compass" in $\Re^{1,1}$. The colored lines show equal-interval contours with respect to the origin; (c) All possible embeddings in $\Re^{2,1}$ (resp. $\Re^3$) are mapped to a sub-manifold of $\Delta_n$, as shown by the red (resp. blue) line. Dimensionality reduction projects the input $\boldsymbol{p}^\star$ onto these sub-manifolds, e. g. by minimizing the KL divergence.

## 3 The representation capability of space-time

This section formally discusses some basic properties of $\Re^{d_s,d_t}$ in relation to dimensionality reduction. We first build a tool to shift between two different representations of an embedding: a matrix of $c(\boldsymbol{y}_i, \boldsymbol{y}_j)$ and a matrix of $\langle \boldsymbol{y}_i, \boldsymbol{y}_j \rangle$. From straightforward derivations, we have

**Lemma 1.** $\mathscr{C}_n = \{C_{n \times n} : \forall i, C_{ii} = 0; \forall i \neq j, C_{ij} = C_{ji}\}$ and $\mathscr{K}_n = \{K_{n \times n} : \forall i, \sum_{j=1}^n K_{ij} = 0; \forall i \neq j, K_{ij} = K_{ji}\}$ are two families of real symmetric matrices. $\dim(\mathscr{C}_n) = \dim(\mathscr{K}_n) = n(n-1)/2$. A linear mapping from $\mathscr{C}_n$ to $\mathscr{K}_n$ and its inverse are given by

$$\boldsymbol{K}(\boldsymbol{C}) = -\frac{1}{2}(\boldsymbol{I}_n - \frac{1}{n}\boldsymbol{e}\boldsymbol{e}^T)\boldsymbol{C}(\boldsymbol{I}_n - \frac{1}{n}\boldsymbol{e}\boldsymbol{e}^T), \quad \boldsymbol{C}(\boldsymbol{K}) = \mathtt{diag}(\boldsymbol{K})\boldsymbol{e}^T + \boldsymbol{e}\mathtt{diag}(\boldsymbol{K})^T - 2\boldsymbol{K}, \quad (4)$$

where $\boldsymbol{e} = (1, \cdots, 1)^T$, and $\mathtt{diag}(\boldsymbol{K})$ means the diagonal entries of $\boldsymbol{K}$ as a column vector.

$\mathscr{C}_n$ and $\mathscr{K}_n$ are the sets of interval matrices and "pseudo-Gram" matrices, respectively [3, 12]. In particular, a p. s. d. $\boldsymbol{K} \in \mathscr{K}_n$ means a Gram matrix, and the corresponding $\boldsymbol{C}(\boldsymbol{K})$ means a square distance matrix. The double centering mapping $\boldsymbol{K}(\boldsymbol{C})$ is widely used to generate a (pseudo-)Gram matrix from a dissimilarity matrix.

**Proposition 2.** $\forall \boldsymbol{C}^\star \in \mathscr{C}_n$, $\exists n$ events in $\Re^{d_s,d_t}$, s. t. $d_s + d_t \leq n-1$ and their intervals are $\boldsymbol{C}^\star$.

*Proof.* $\forall \boldsymbol{C}^\star \in \mathscr{C}_n$, $\boldsymbol{K}^\star = \boldsymbol{K}(\boldsymbol{C}^\star)$ has the eigen-decomposition $\boldsymbol{K}^\star = \sum_{l=1}^{\mathtt{rank}(\boldsymbol{K}^\star)} \lambda_l^\star \boldsymbol{v}_l^\star (\boldsymbol{v}_l^\star)^T$ where $\mathtt{rank}(\boldsymbol{K}^\star) \leq n-1$ and $\{\boldsymbol{v}_l^\star\}$ are orthonormal. For each $l = 1, \cdots, \mathtt{rank}(\boldsymbol{K}^\star)$, $\sqrt{|\lambda_l^\star|}\boldsymbol{v}_l^\star$ gives the coordinates in one dimension, which is space-like if $\lambda_l^\star > 0$ or time-like if $\lambda_l^\star < 0$. $\quad\square$

**Remark 2.1.** $\Re^{d_s,d_t}$ $(d_s + d_t \leq n-1)$ can represent any interval matrix $\boldsymbol{C}^\star \in \mathscr{C}_n$, or equivalently, any $\boldsymbol{K}^\star \in \mathscr{K}_n$. Comparatively, $\Re^d$ $(d \leq n-1)$ can only represent $\{\boldsymbol{K} \in \mathscr{K}_n : \boldsymbol{K} \succeq 0\}$.

A pair-wise distance matrix in $\Re^d$ is invariant to rotations. In other words, the direction information of a point cloud is completely discarded. In $\Re^{d_s,d_t}$, some direction information is kept to distinguish between space-like and time-like dimensions. As shown in fig. 1b, one can tell the direction in $\Re^{1,1}$ by moving a point along the curve $\{(y^1)^2 + (y^2)^2 = 1\}$ and measuring its interval w. r. t. the origin.

Local embedding techniques often use similarity measures in a statistical simplex $\Delta_n = \left\{ \boldsymbol{p} = (p_{ij}) : 1 \leq i \leq n; 1 \leq j \leq n; i < j; \forall i, \forall j, p_{ij} > 0; \sum_{i,j:i<j} p_{ij} = 1 \right\}$. This $\Delta_n$ has one less dimension than $\mathscr{C}_n$ and $\mathscr{K}_n$ so that $\dim(\Delta_n) = n(n-1)/2 - 1$. A mapping from $\mathscr{K}_n$ ($\mathscr{C}_n$) to $\Delta_n$ is given by

$$p_{ij} \propto f\left(\boldsymbol{C}_{ij}(\boldsymbol{K})\right), \quad (5)$$

where $f(\cdot)$ is a positive-valued strictly monotonically decreasing function, so that a large probability mass is assigned to a pair of events with a small interval. Proposition 2 trivially extends to

**Proposition 3.** $\forall \boldsymbol{p}^\star \in \Delta_n$, $\exists n$ events in $\Re^{d_s,d_t}$, s. t. $d_s + d_t \leq n-1$ and their similarities are $\boldsymbol{p}^\star$.

**Remark 3.1.** $\Re^{d_s,d_t}$ $(d_s + d_t \leq n-1)$ can represent any $n \times n$ symmetric positive similarities.

Typically in eq. (5) we have $f(x) = \exp(-x)$. The pre-image in $\mathscr{C}_n$ of any given $\boldsymbol{p}^\star \in \Delta_n$ is the curve $\{\boldsymbol{C}^\star + 2\delta\left(\boldsymbol{e}\boldsymbol{e}^T - \boldsymbol{I}_n\right) : \forall i \neq j, C_{ij}^\star = -\ln p_{ij}^\star; \delta \in \Re\}$, where $2\delta\left(\boldsymbol{e}\boldsymbol{e}^T - \boldsymbol{I}_n\right)$ means a uniform increment on the off-diagonal entries of $\boldsymbol{C}^\star$. By eq. (4), the corresponding curve in $\mathscr{K}_n$ is $\{\boldsymbol{K}^\star(\delta) = \boldsymbol{K}^\star + \delta\left(\boldsymbol{I}_n - \frac{1}{n}\boldsymbol{e}\boldsymbol{e}^T\right) : \delta \in \Re\}$, where $\boldsymbol{K}^\star(0) = \boldsymbol{K}^\star = \boldsymbol{K}(\boldsymbol{C}^\star)$. Because $\left(\boldsymbol{I}_n - \frac{1}{n}\boldsymbol{e}\boldsymbol{e}^T\right)$ shares with $\boldsymbol{K}^\star$ a common eigenvector $\boldsymbol{e}$ with zero eigenvalue, and the rest eigenvalues are all 1, there exist orthonormal vectors $\{\boldsymbol{v}_l^\star\}_{l=1}^{n-1}$ and real numbers $\{\lambda_l^\star\}_{l=1}^{\text{rank}(\boldsymbol{K}^\star)}$, s. t. $\boldsymbol{K}^\star = \sum_{l=1}^{\text{rank}(\boldsymbol{K}^\star)} \lambda_l^\star \boldsymbol{v}_l^\star (\boldsymbol{v}_l^\star)^T$, and $\left(\boldsymbol{I}_n - \frac{1}{n}\boldsymbol{e}\boldsymbol{e}^T\right) = \sum_{l=1}^{n-1} \boldsymbol{v}_l^\star(\boldsymbol{v}_l^\star)^T$. Therefore

$$\boldsymbol{K}^\star(\delta) = \sum_{l=1}^{\text{rank}(\boldsymbol{K}^\star)} (\lambda_l^\star + \delta)\boldsymbol{v}_l^\star(\boldsymbol{v}_l^\star)^T + \sum_{l=\text{rank}(\boldsymbol{K}^\star)+1}^{n-1} \delta\boldsymbol{v}_l^\star(\boldsymbol{v}_l^\star)^T. \qquad (6)$$

Depending on $\delta$, $\boldsymbol{K}^\star(\delta)$ can be negative definite, positive definite, or somewhere in between. This is summarized in the following theorem.

**Theorem 4.** *If $f(x) = \exp(-x)$ in eq. (5), the pre-image in $\mathscr{K}_n$ of $\forall \boldsymbol{p}^\star \in \Delta_n$ is a continuous curve $\{\boldsymbol{K}^\star(\delta) : \delta \in \Re\}$. $\exists \delta_0, \delta_1 \in \Re$, s. t. $\forall \delta < \delta_0$, $\boldsymbol{K}^\star(\delta) \prec 0$, $\forall \delta > \delta_1$, $\boldsymbol{K}^\star(\delta) \succ 0$, and the number of positive eigenvalues of $\boldsymbol{K}^\star(\delta)$ increases monotonically with $\delta$.*

With enough dimensions, any $\boldsymbol{p}^\star \in \Delta_n$ can be perfectly represented in a space-only, or time-only, or space-time-mixed $\Re^{d_s,d_t}$. There is no particular reason to favor a space-only model, because the objective of dimensionality reduction is to get a compact model with a small number of dimensions, regardless of whether they are space-like or time-like. Formally, $\mathscr{K}_n^{d_s,d_t} = \{\boldsymbol{K}^+ - \boldsymbol{K}^- : \text{rank}(\boldsymbol{K}^+) \leq d_s; \text{rank}(\boldsymbol{K}^-) \leq d_t; \boldsymbol{K}^+ \succeq 0; \boldsymbol{K}^- \succeq 0\}$ is a low-rank subset of $\mathscr{K}_n$. In the domain $\mathscr{K}_n$, dimensionality reduction based on the input $\boldsymbol{p}^\star$ finds some $\hat{\boldsymbol{K}}^{d_s,d_t} \in \mathscr{K}_n^{d_s,d_t}$, which is close to the curve $\boldsymbol{K}^\star(\delta)$.

In the probability domain $\Delta_n$, the image of $\mathscr{K}_n^{d_s,d_t}$ under some mapping $g : \mathscr{K}_n \to \Delta_n$ is $g(\mathscr{K}_n^{d_s,d_t})$. As shown in fig. 1c, dimensionality reduction finds some $\hat{\boldsymbol{p}}^{d_s,d_t} \in g(\mathscr{K}_n^{d_s,d_t})$, so that $\hat{\boldsymbol{p}}^{d_s,d_t}$ is the closest point to $\boldsymbol{p}^\star$ w. r. t. some information theoretic measure. The proximity of $\boldsymbol{p}^\star$ to $\hat{\boldsymbol{p}}^{d_s,d_t}$, i. e. its proximity to $g(\mathscr{K}_n^{d_s,d_t})$, measures the quality of the model $\Re^{d_s,d_t}$ as the embedding target space, when the model scale or the number of dimensions is given.

We will investigate the latter approach, which depends on the choice of $d_s$, $d_t$, the mapping $g$, and some proximity measure on $\Delta_n$. We will show that, with the same number of dimensions $d_s + d_t$, the region $g(\mathscr{K}_n^{d_s,d_t})$ with space-time-mixed dimensions is naturally close to certain input $\boldsymbol{p}^\star$.

## 4  Space-time local embeddings

We project a given similarity matrix $\boldsymbol{p}^\star \in \Delta_n$ to some $\hat{\boldsymbol{K}} \in \mathscr{K}_n^{d_s,d_t}$, or equivalently, to a set of events $\boldsymbol{Y} = \{\boldsymbol{y}_i\}_{i=1}^n \subset \Re^{d_s,d_t}$, so that $\forall i$, $\forall j$, $\langle \boldsymbol{y}_i, \boldsymbol{y}_j \rangle = \hat{K}_{ij}$ as in eq. (2), and the similarities among these events resemble $\boldsymbol{p}^\star$. As discussed in section 3, a mapping $g : \mathscr{K}_n \to \Delta_n$ helps transfer $\mathscr{K}_n^{d_s,d_t}$ into a sub-manifold of $\Delta_n$, so that the projection can be done inside $\Delta_n$. This mapping expressed in the event coordinates is given by

$$p_{ij}(\boldsymbol{Y}) \propto \frac{\exp\left(\|\boldsymbol{y}_i^t - \boldsymbol{y}_j^t\|^2\right)}{1 + \|\boldsymbol{y}_i^s - \boldsymbol{y}_j^s\|^2}, \qquad (7)$$

where $\boldsymbol{y}^s = (y^1, \ldots, y^{d_s})^T$, $\boldsymbol{y}^t = (y^{d_s+1}, \ldots, y^{d_s+d_t})^T$, and $\|\cdot\|$ denotes the 2-norm. For any pair of events $\boldsymbol{y}_i$ and $\boldsymbol{y}_j$, $p_{ij}(\boldsymbol{Y})$ increases when their space coordinates move close, and/or when their time coordinates move away. This agrees with the basic intuitions of space-time. For time-like dimensions, the heat kernel is used to make $p_{ij}(\boldsymbol{Y})$ sensitive to time variations. This helps to suppress events with large absolute time values, which make the embedding less interpretable. For space-like dimensions, the Student-$t$ kernel, as suggested by t-SNE [13], is used, so that there could be more "volume" to accommodate the often high-dimensional input data. Based on our experience, this hybrid parametrization of $p_{ij}(\boldsymbol{Y})$ can better model real data as compared to alternative parametrizations. Similar to SNE [4] and t-SNE [13], an optimal embedding can be obtained by minimizing the Kullback-Leibler (KL) divergence from the input $\boldsymbol{p}^\star$ to the output $p(\boldsymbol{Y})$, given by

$$\text{KL}(\boldsymbol{Y}) = \sum_{i,j:i<j} p_{ij}^\star \ln \frac{p_{ij}^\star}{p_{ij}(\boldsymbol{Y})}. \qquad (8)$$

According to some straightforward derivations, its gradients are

$$\frac{\partial \text{KL}}{\partial \boldsymbol{y}_i^t} = -2 \sum_{j:j \neq i} \left( p_{ij}^\star - p_{ij}(\boldsymbol{Y}) \right) \left( \boldsymbol{y}_i^t - \boldsymbol{y}_j^t \right), \tag{9}$$

$$\frac{\partial \text{KL}}{\partial \boldsymbol{y}_i^s} = 2 \sum_{j:j \neq i} \frac{1}{1 + \|\boldsymbol{y}_i^s - \boldsymbol{y}_j^s\|^2} \left( p_{ij}^\star - p_{ij}(\boldsymbol{Y}) \right) \left( \boldsymbol{y}_i^s - \boldsymbol{y}_j^s \right), \tag{10}$$

where $\forall i$, $\forall j$, $p_{ij}^\star = p_{ji}^\star$ and $p_{ij}(\boldsymbol{Y}) = p_{ji}(\boldsymbol{Y})$. As an intuitive interpretation of a gradient descent process w. r. t. eqs. (9) and (10), we have that if $p_{ij}(\boldsymbol{Y}) < p_{ij}^\star$, i. e. $\boldsymbol{y}_i$ and $\boldsymbol{y}_j$ are put too far from each other, then $\boldsymbol{y}_i^s$ and $\boldsymbol{y}_j^s$ are attracting, and $\boldsymbol{y}_i^t$ and $\boldsymbol{y}_j^t$ are repelling, so that their space-time interval becomes shorter; if $p_{ij}(\boldsymbol{Y}) > p_{ij}^\star$, then $\boldsymbol{y}_i$ and $\boldsymbol{y}_j$ are repelling in space and attracting in time.

During gradient descent, $\{\boldsymbol{y}_i^s\}$ are updated by the delta-bar-delta scheme as used in t-SNE [13], where each scalar parameter has its own adaptive learning rate initialized to $\gamma^s > 0$; $\{\boldsymbol{y}_i^t\}$ are updated based on one global adaptive learning rate initialized to $\gamma^t > 0$. The learning of time should be more cautious, because $p_{ij}(\boldsymbol{Y})$ is more sensitive to time variations by eq. (7). Therefore, the ratio $\gamma^t / \gamma^s$ should be very small, e.g. $1/100$.

## 5  Empirical results

Aiming at potential applications in data visualization and social network analysis, we compare SNE [4], t-SNE [13], and the method proposed in section 4 denoted as SNE$^\text{ST}$ . They are based on the same optimizer but correspond to different sub-manifolds of $\Delta_n$, as presented by the curves in fig. 1c. Given different embeddings of the same dataset using the same number of dimensions, we perform model selection based on the KL divergence as explained in the end of section 3.

We generated a toy dataset SCHOOL, representing a school with two classes. Each class has 20 students standing evenly on a circle, where each student is communicating with his (her) 4 nearest neighbours, and one teacher, who is communicating with all the students in the same class and the teacher in the other class. The input $\boldsymbol{p}^\star$ is distributed evenly on the pairs $(i, j)$ who are socially connected.

NIPS22 contains a $4197 \times 3624$ author-document matrix from NIPS 1988 to 2009 [2]. After discarding the authors who have only one NIPS paper, we get 1418 authors who co-authored 2121 papers. The co-authorship matrix is $\text{CA}_{1418 \times 1418}$, where $\text{CA}_{ij}$ denotes the number of papers that author $i$ co-authored with author $j$. The input similarity $\boldsymbol{p}^\star$ is computed so that $p_{ij}^\star \propto \text{CA}_{ij}(1/\sum_j \text{CA}_{ij} + 1/\sum_i \text{CA}_{ij})$, where the number of co-authored papers is normalized by each author's total number of papers. NIPS17 is built in the same way using only the first 17 volumes.

GrQc is an arXiv co-authorship graph [16] with 5242 nodes and 14496 edges. After removing one isolated node, a matrix $\text{CA}_{5241 \times 5241}$ gives the numbers of co-authored papers between any two authors who submitted to the general relativity and quantum cosmology category from January 1993 to April 2003. The input similarity $\boldsymbol{p}^\star$ satisfies $p_{ij}^\star \propto \text{CA}_{ij}(1/\sum_j \text{CA}_{ij} + 1/\sum_i \text{CA}_{ij})$.

W5000 is the semantic similarities among 5000 English words in $\text{WS}_{5000 \times 5000}$ [2, 17]. Each $\text{WS}_{ij}$ is an asymmetric non-negative similarity from word $i$ to word $j$. The input is normalized into a probability vector $\boldsymbol{p}^\star$ so that $p_{ij}^\star \propto \text{WS}_{ij}/\sum_j \text{WS}_{ij} + \text{WS}_{ji}/\sum_i \text{WS}_{ji}$. W1000 is built in the same way using a subset of 1000 words.

Table 1 shows the KL divergence in eq. (8). In most cases, SNE$^\text{ST}$ for a fixed number of free parameters has the lowest KL. On NIPS22, GrQc, W1000 and W5000, the embedding by SNE$^\text{ST}$ in $\Re^{2,1}$ is even better than SNE and t-SNE in $\Re^4$, meaning that the embedding by SNE$^\text{ST}$ is both compact and faithful. This is in contrast to the mixture approach for visualization [2], which multiplies the number of parameters to get a faithful representation.

Fixing the free parameters to two dimensions, t-SNE in $\Re^2$ has the best overall performance, and SNE$^\text{ST}$ in $\Re^{1,1}$ is worse. We also discovered that, using $d$ dimensions, $\Re^{d-1,1}$ usually performs better than alternative choices such as $\Re^{d-2,2}$, which are not shown due to space limitation. A time-like dimension allows adaptation to non-metric data. The investigated similarities, however, are

Table 1: KL divergence of different embeddings. After repeated runs on different configurations for each embedding, the minimal KL that we have achieved within 5000 epochs is shown. The bold numbers show the winners among SNE, t-SNE and SNE$^{\text{ST}}$ using the same number of parameters.

| | SCHOOL | NIPS17 | NIPS22 | GrQc | W1000 | W5000 |
|---|---|---|---|---|---|---|
| SNE $\to \Re^2$ | 0.52 | 1.88 | 2.98 | 3.19 | 3.67 | 4.93 |
| SNE $\to \Re^3$ | 0.36 | 0.85 | 1.79 | 1.82 | 3.20 | 4.42 |
| SNE $\to \Re^4$ | **0.19** | **0.35** | 1.01 | 1.03 | 2.76 | 3.93 |
| t-SNE $\to \Re^2$ | 0.61 | **0.88** | **1.29** | **1.24** | **2.15** | **3.00** |
| t-SNE $\to \Re^3$ | 0.58 | 0.85 | 1.23 | 1.14 | 2.00 | 2.79 |
| t-SNE $\to \Re^4$ | 0.58 | 0.84 | 1.22 | 1.11 | 1.96 | 2.74 |
| SNE$^{\text{ST}} \to \Re^{1,1}$ | **0.43** | 0.91 | 1.62 | 2.34 | 2.59 | 3.64 |
| SNE$^{\text{ST}} \to \Re^{2,1}$ | **0.31** | **0.60** | **0.97** | **1.00** | **1.92** | **2.57** |
| SNE$^{\text{ST}} \to \Re^{3,1}$ | 0.29 | 0.54 | **0.93** | **0.88** | **1.79** | **2.39** |

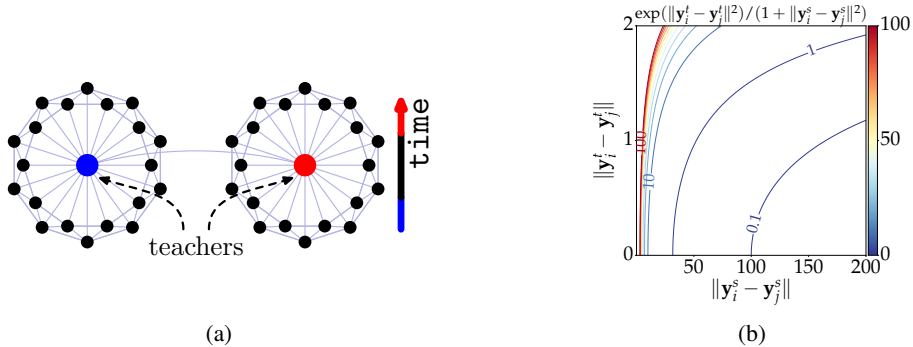

(a)                    (b)

Figure 2: (a) The embedding of SCHOOL by SNE$^{\text{ST}}$ in $\Re^{2,1}$. The black (resp. colored) dots denote the students (resp. teachers). The paper coordinates (resp. color) mean the space (resp. time) coordinates. The links mean social connections. (b) The contour of $\frac{\exp\left(\|\boldsymbol{y}_i^t - \boldsymbol{y}_j^t\|^2\right)}{1 + \|\boldsymbol{y}_i^s - \boldsymbol{y}_j^s\|^2}$ in eq. (7) as a function of $\|\boldsymbol{y}_i^s - \boldsymbol{y}_j^s\|$ (x-axis) and $\|\boldsymbol{y}_i^t - \boldsymbol{y}_j^t\|$ (y-axis). The unit of the displayed levels is $10^{-3}$.

mainly space-like, in the sense that a random pair of people or words are more likely to be dissimilar (space-like) rather than similar (time-like). According to our experience, on such datasets, good performance is often achieved with mainly space-like dimensions mixed with a small number of time-dimensions, e.g. $\Re^{2,1}$ or $\Re^{3,1}$ as suggested by table 1.

To interpret the embeddings, fig. 2a presents the embedding of SCHOOL in $\Re^{2,1}$, where the space and time are represented by paper coordinates and three colors levels, respectively. Each class is embedded as a circle. The center of each class, the teacher, is lifted to a different time, so as to be near to all students in the same class. One teacher being blue, while the other being red, creates a "hyper-link" between the teachers, because their large time difference makes them nearby in $\Re^{2,1}$.

Figures 3 and 4 show the embeddings of NIPS22 and W5000 in $\Re^{2,1}$. Similar to the (t-)SNE visualizations [2, 4, 13], it is easy to find close authors or words embedded nearby. The learned $p(\boldsymbol{Y})$, however, is not equivalent to the visual proximity, because of the counter-intuitive time dimension. *How much does the visual proximity reflect the underlying $p(\boldsymbol{Y})$?* From the histogram of the time coordinates, we see that the time values are in the narrow range $[-1.5, 1.5]$, while the range of the space coordinates is at least 100 times larger. Figure 2b shows the similarity function on the right-hand-side of eq. (7) over an interesting range of $\|\boldsymbol{y}_i^s - \boldsymbol{y}_j^s\|$ and $\|\boldsymbol{y}_i^t - \boldsymbol{y}_j^t\|$. In this range, large similarity values are very sensitive to space variations, and their red level curves are almost vertical, meaning that the similarity information is largely carried by space coordinates. Therefore, the visualization of neighborhoods is relatively accurate: visually nearby points are indeed similar; proximity in a neighborhood is *informative* regarding $p(\boldsymbol{Y})$. On the other hand, small similarity values are less sensitive to space variations, and their blue level curves span a large distance in space, meaning that the visual distance between dissimilar points is less informative regarding $p(\boldsymbol{Y})$. For

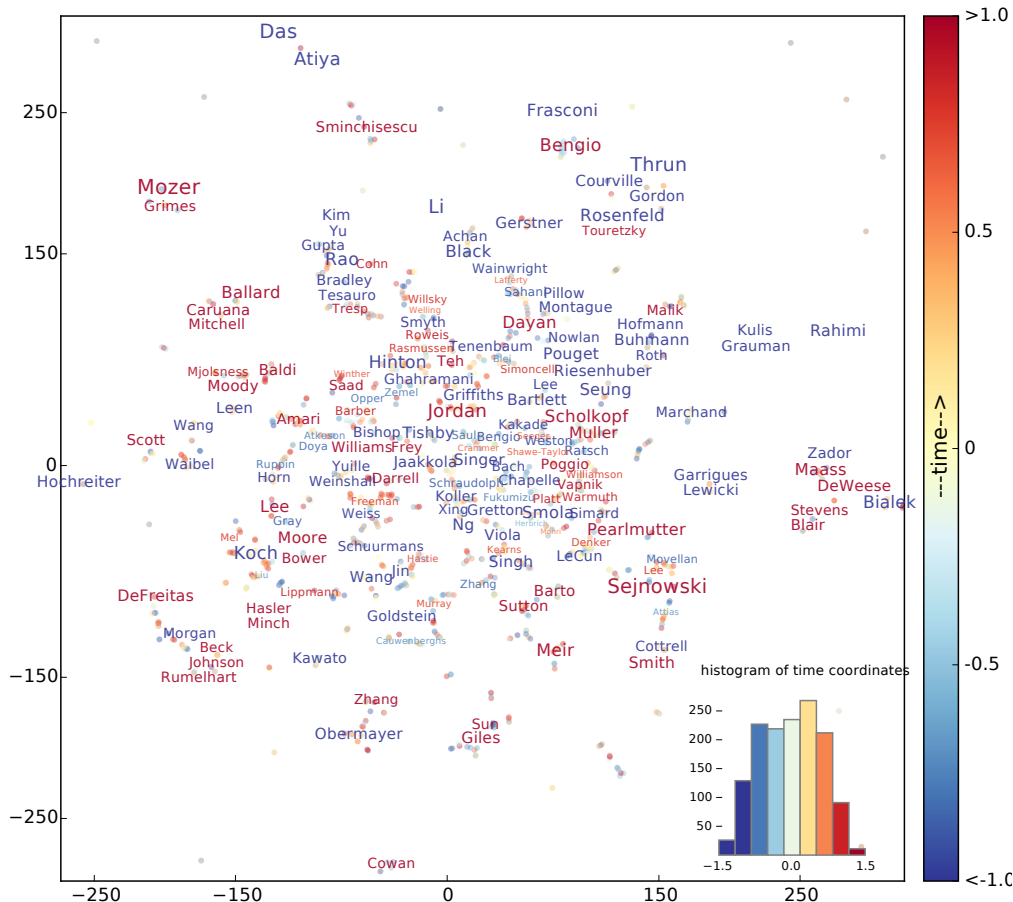

Figure 3: An embedding of `NIPS22` in $\Re^{2,1}$. "Major authors" with at least 10 NIPS papers or with a time value in the range $(-\infty, -1] \cup [1, \infty)$ are shown by their names. Other authors are shown by small dots. The paper coordinates are in space-like dimensions. The positions of the displayed names are adjusted up to a tiny radius to avoid text overlap. The color of each name represents the time dimension. The font size is proportional to the absolute time value.

example, a visual distance of 165 with a time difference of 1 has roughly the same similarity as a visual distance of 100 with no time difference. This is a matter of embedding dissimilar samples far or very far and does not affect much the visual perception, which naturally requires less accuracy on such samples. However, perception errors could still occur in these plots, although they are increasingly unlikely as the observation radius turns small. In viewing such visualizations, one must *count in the time* represented by the colors and font sizes, and remember that a point with a large absolute time value should be weighted higher in similarity judgment.

Consider the learning of $\boldsymbol{y}_i$ by eq. (9), if the input $p_{ij}^\star$ is larger than what can be faithfully modeled in a space-only model, then $j$ will push $i$ to a different time. Therefore, the absolute value of time is a *significance* measurement. By fig. 2a, the connection hubs, and points with remote connections, are more likely to be at a different time. Emphasizing the embedding points with large absolute time values helps the user to focus on important points. One can easily identify well-known authors and popular words in figs. 3 and 4. This type of information is not discovered by traditional embeddings.

## 6 Conclusions and Discussions

We advocate the use of space-time representation for non-metric data. While previous works on such embeddings [3, 12] compute an indefinite kernel by simple transformations of the input data, we learn a low-rank indefinite kernel by manifold learning, trying to better preserve the neigh-

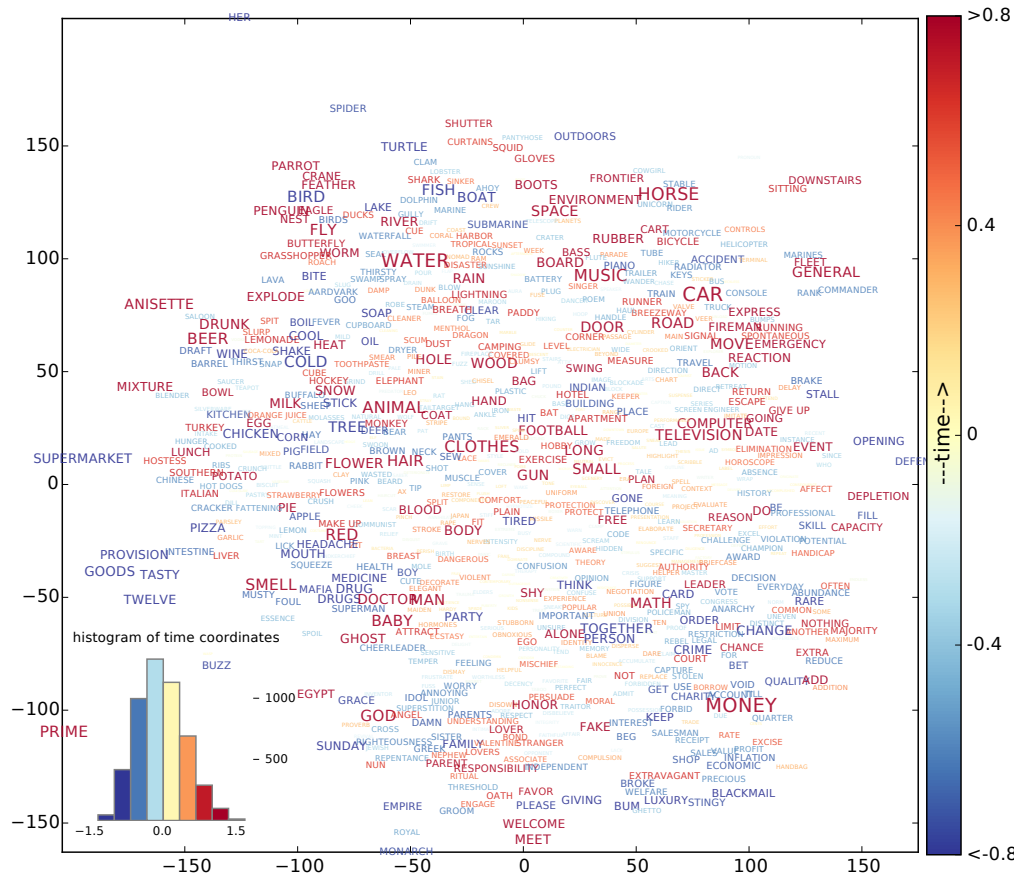

Figure 4: An embedding of W5000 in $\Re^{2,1}$. Only a subset is shown for a clear visualization. The position of each word represents its space coordinates up to tiny adjustments to avoid overlap. The color of each word shows its time value. The font size represents the absolute time value.

bours [4]. We discovered that, using the same number of dimensions, certain input information is better preserved in space-time than Euclidean space. We built a space-time visualizer of non-metric data, which automatically discovers important points.

To enhance the proposed visualization, an interactive interface can allow the user select one reference point, and show the true similarity values, e.g., by aligning other points so that the visual distances correspond to the similarities. Proper constraints or regularization could be proposed, so that the time values are discrete or sparse, and the resulting embedding can be more easily interpreted.

The proposed learning is on a sub-manifold $\mathscr{K}_n^{d_s,d_t} \subset \mathscr{K}_n$, or a corresponding sub-manifold of $\Delta_n$. Another interesting sub-manifold of $\mathscr{K}_n$ could be $\{ \boldsymbol{K} - \boldsymbol{t}\boldsymbol{t}^T \ : \ \boldsymbol{K} \succ 0; \ \boldsymbol{t} \in \Re^n \}$, which extends the p. s. d. cone to any matrix in $\mathscr{K}_n$ with a compact negative eigen-spectrum. It is possible to construct a sub-manifold of $\mathscr{K}_n$ so that the embedder can learn whether a dimension is space-like or time-like.

As another axis of future investigation, given the large family of manifold learners, there can be many ways to project the input information onto these sub-manifolds. The proposed method SNE[ST] is based on the KL divergence in $\Delta_n$. Some immediate extensions can be based on other dissimilarity measures in $\mathscr{K}_n$ or $\Delta_n$. This could also be useful for faithful representations of graph datasets with indefinite weights.

## Acknowledgments

This work has been supported be the Department of Computer Science, University of Geneva, in collaboration with Swiss National Science Foundation Project MAAYA (Grant number 144238).

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
