[Reviews · NeurIPS 2015]

Submitted by Assigned_Reviewer_1

====== After reading their rebuttal ==========

Thanks for answering all my questions. I will keep my rating the same (Good paper, accept 7).
Summary: The paper presents a data visualisation method based on the concept of space-time. The space-time representation is capable of showing a broader family of proximities than an Euclidean space with the same dimensionality. Based on the KL measure, the authors argue that the lower dimensional representation of the high dimensional data using the space-time local embedding method can keep more information than Euclidean embeddings. I am quite convinced, but there is one question about interpretability of the visualised data in space-time.

Other than lower KL divergence and those mentioned in the two paragraphs in page 7, can authors articulate possible ways of interpreting their visualisation? What could that time-like dimension mean? What does the space-like dimension correspond to?

Submitted by Assigned_Reviewer_2

This paper proposes the use of space-time (indefinite) metric in dimension reduction and demonstrate its usage in SNE. The idea is nice and intuitive.

Concerns: It is not easy to see whether the quality of embedding is much better than using positive definite metrics from the figures, and the interpretation of the embedding is a bit non-intuitive as the authors also pointed out; I imagine it is even harder when higher dimensional (>3) embeddings is computed. Also since the authors claim it is easier for space-time embedding to capture information with the same amount of parameters, I wonder if the embedding is more useful for some supervised or semi-supervised tasks than "positive definite embeddings".
Summary: This paper proposes the idea of space-time (indefinite) metric in dimension reduction. The idea is nice and intuitive but I would appreciate more evidence of its advantages over existing embeddings.

Submitted by Assigned_Reviewer_3

This paper proposes a new embedding structure for the output of nonlinear dimensionality reduction (NLDR). It borrows the space-time concept from physics and shows that the new structure can encode (or represent) more information than conventional Euclidean space.

The idea is pioneering and wonderful. It goes beyond all conventional embedding structures and has theoretical guarantee to accommodate more symmetry in the input space. It brings much flexibility to the NLDR problem. This can be an important middle step towards better dimensionality reduction.

However, how to visualize the space-time structure seems unsolved. 1) The authors proposes to use colors to show the time dimension. But it is pretty hard to read such display. Information visualization should take advantage of the human eye's broad bandwidth pathway into the mind to allow users to digest large amounts of information at once. But Figs. 1 and 2 are rather messy. There is no immediate and clear macro pattern in these figures. Users have to examine the details piece by piece, which is time-consuming.

2) the meaning of the time dimension in data science remains unclear in many aspects. For example, in Fig. 1, what is the different meanings of "very red" and "very green" (e.g. Hinton vs. Bengio)? In Fig. 2, what are the meaning of the red groups and blue groups?

The KL-divergence is used as benchmarks to show the advantage of using space-time. This is not convincing because there are many other factors that can affect the results. For example, the benefits may vanish when another input kernel width or another optimization algorithm is used. In addition, the SNE and t-SNE objectives are not scale invariant to Y. That is, multiplying a same constant to all y's may change the objective. A smaller divergence may not bring a better visualization. So I don't agree Table 1 is a solid justification.

The title is a bit ambiguous. Readers may think this is to find embeddings of spatial-temporal input data. How about "Local Embeddings using Space-Time"?

Line 341: what do you mean "an author or a word A at a different time with the majority is close to them all"?

Line 346: what do you mean "dark color"?

After feedback: The author feedback is a bit disappointing. Repeatedly arguing the advantages is not very helpful. I believe the idea is new and interesting. It deserves further exploration. But the visualization part needs improvement. From the feedback, I feel that visualizing the information in a space-time is unsolved. I don't agree with the 3D visualization approach because 1) it is restricted to only R^{2,1} case and 2) it is not provided in the paper or supplemental material. Using colors or levels to show the time dimensional is not satisfied either. The authors admit that the visualization work is still ongoing. The time dimension is claimed to show the importance (or ranking) of data points. But this remains unclear in Figure 2. The authors did not answer my question about the meanings of red words and blue words. Furthermore, it seems only the absolute values of time dimension is important. We still don't know the meaning with the signs. For example, Hinton and LeCun are green (blue in the final version), but Bengio is very red. Why? The feedback cannot explain this. The authors admit that selecting the right evaluation measure for visualization is an open problem. Therefore smaller KL-divergence values only partially demonstrate the advantages. It is not "a solid justification" as claimed. As I said, the reduced KL is not necessarily caused by the use of space-time and it may not correspond to better displays.

After seeing the final version: The authors have removed most improper claims in the original version. They also improved the quality of the paper. But most of the above questions are not answered yet. They deserve further studies in future research.
Summary: A pioneering work of using a new output embedding structure for nonlinear dimensionality reduction. The visualization part needs improvement.

Submitted by Assigned_Reviewer_4

The paper describes a dimensionality reduction method whose target space is not a Euclidean space, but rather a Minkowski space (the eponymous space-time). The motivation for looking at this broader class of spaces is to allow a broader class of similarities (e.g., waving the requirement that the similarity matrix be positive semi-definite). One practical motivation for this is to allow an arbitrarily large number of points to share a common nearest neighbor in the low-dimensional space.

The authors describe an algorithm that finds such an embedding: essentially it's a gradient descent method that minimizes Kullback-Leibler divergence between the input and output similarities. The authors present experiments to illustrate the value of this embedding and to motivate their arguments in its favor.

I am torn about the paper. On the one hand, the idea is interesting, and some of the experiments indeed support its main premise; on the other hand, the visualization experiments to me are a strong argument against the idea. (To put it differently, I think I would give the paper a higher score if the authors framed it differently experimentally. See details in comments.)

* The lower KL divergences of the embeddings are a good argument for this

method, but then you need something else, some method that uses the embedded

points to make the argument for usefulness, because when it comes to

visualization, the lower KL divergence does not nearly make up for the

unintuitive space-time embedding.

* The statement "The space-time visualization is not intuitive in the

traditional sense" is a major understatement. Even with the offered intuition

that points with high absolute time value are close to other points, I find

it very difficult to make sense of the presented embeddings. It's difficult

to judge how much of it is due to the choices that the authors make (e.g.,

clipping the time values where to show colors) and how much of it is the

problem with visualizing space-time itself. I have a feeling the problem is

with the target space itself.

To me, these visualizations are an argument against this technique -- they

seem hopeless as a data analysis tool.

I realize this is a unfair to the

larger idea of space-time embeddings, which are perfectly interesting for

other reasons.

* It seems to me that this method could be very useful for producing some kind

of an importance ranking of the input points by their (absolute) time value.

You sort of try to make this argument on specific examples in Figures 1 and

2, and allude to it at the end of Section 4, but that's not enough in my

opinion. It would be much better to expand and formalize this argument,

forgetting about the actual visualization, and to relate (even if

empirically) this ranking to some other way to rank importance (either in

social graphs or on some other data set).

* Minor comment: Proposition 4 is a strange way to answer the preceding

questions. What does a random matrix A have to do with reality? Reality is

patently different from noise. Without this assumption, dimensionality

reduction research would have a very hard time.
Summary: The idea behind the paper is interesting, but the presented experiments argue against the technique as much as they do in its favor. I think the paper would be stronger if it was framed using different types of experiments.

Author Feedback
Author rebuttal: We thank all reviewers for the comments. We address the main points as follows.

-- usefulness of the visualization

The histogram of the time values presents an exponential shape: the percentage of embedding points decreases as the absolute time-value increases (this is not mentioned but will be added). Moreover, the scale of time is much smaller than the scale of space (see fig.2-3) Therefore, the main parts of the visualization is still intuitive, because the large space distance will compensate the counter-intuitions caused by time. For example, in fig.3, it is clear that similar words are grouped together.

Using 3D visualization technologies, we can just plot the space-time as it is, using a horizontal plane to present time 0; above this plane is time>0; below is time<0. To enhance the immediate perception rate, each embedding point can be visually linked to all neighbors within its light-cone (fig.1), meaning that the linked points have a "distance" smaller than 0 (or epsilon). Points without (or with little) such links can be regarded as traditional Euclidean embedding points. Points with many such links are counter-intuitive points, which can be regarded as the center of a group of points.

On a 2D paper or screen, one way to enhance the intuition could be discretizing the learned time values into several levels (e.g.,-2,-1,0,1,2), visualizing the discretized values instead, and linking each embedding point to its \epsilon neighbours (as in the above 3D solution). The idea is to emphasize a small percent of counter-intuitive points. Another way to gain intuition is to impose constraints/regularizers to the embedder, e.g., forcing the embedding points to be sparse (or discrete) in time. We are tunning the visualizations and will include the best solution.

Remarkably, the proposed visualization reveals new information that is not presented in traditional embeddings: one can look at the time dimension and identify the "significant" points (see following)

-- the meaning of the time dimension

As explained in the last paragraph of sec.4, also pointed out by reviewer 3, the time dimension carries some ranking information. An embedding point with a large absolute time value is considered to be more "significant". This is similar to PageRank, except that the "links" which can cast votes are learned together with an embedding. We can include a more detailed analysis of this ranking mechanism, and possibly some empirical results if space allows.

-- (fig.2) blue authors v.s. red authors

Both the very red authors and very blue authors have many collaborations. A blue author (e.g. LeCun) and a red author (e.g. Bengio) can have a "hyper-link". Along the space dimensions they are likely around the center of their local research groups. One being blue, while the other being red, will enlarge their time difference, and thus shrink their "distance", and thus enlarge their similarities, helping modeling such remote collaborations. This phenomenon, which are common in social networks, cannot be faithfully modeled by an Euclidean embedding.

-- using KL-divergence as benchmark is not convincing

The key benchmark is that a low-dimensional Euclidean embedding simply cannot represent non-metric data or high-dimensional phenomenons, while space-time can.

Consider manifold learning as projecting the input information to a sub-manifold S embedded in the space of all possible embeddings. This paper is proposing a different sub-manifold (rather than a different projection), which is intrinsically close to certain input information.

In limited space, we use a KL-based projection to demonstrate the concept of this sub-manifold. KL is used as the performance metric, because it is the cost function of all the compared methods. KL is not scale-invariant. However the optimizer will select the best scale to achieve the infimum-KL. This is like measuring the distance from a point to a hyper-plane, which is the infimum of all point-to-point distances. The reported KL values measures how far the input information (a point) is from the corresponding sub-manifold S (a curved hyper-plane). We showed that the proposed sub-manifold is closer.

There can be a wide array of projection measures to use, e.g., MDS energy or symmetrized KL. This gives another axis of investigation. If a measure is selected as the embedding cost function, it is very likely that one can obtain a smaller value of these corresponding measures due to the representation power of space-time.

It is a key challenge for manifold learning to select a right evaluation measure. There are many ways, but no universal way, to measure the embedding quality.

-- Could the embedding be useful for (semi-)supervised tasks?

As long as the input proximities have enough non-metric/high-dimensional properties, the space-time representation is expected to give a more faithful model than a classical model with p.s.d. constraints.